# Sensor-Based Assessment of Post-Stroke Shoulder Pain and Balance

**DOI:** 10.3390/s25247665

**Published:** 2025-12-18

**Authors:** Eda Salgut, Gökhan Özkoçak, Arzu Dinç Yavaş

**Affiliations:** 1Department of Physiotherapy and Rehabilitation, Institute of Health Sciences, Istanbul Aydin University, 34295 Istanbul, Turkey; 2Department of Physical Medicine and Rehabilitation, Faculty of Medicine, Istanbul Aydin University, 34295 Istanbul, Turkey; gokhanozkocak@aydin.edu.tr (G.Ö.); arzudincyavas@aydin.edu.tr (A.D.Y.)

**Keywords:** stroke, hemiplegic shoulder pain, balance, wearable sensors, accidental falls, risk assessment

## Abstract

**Background/Objectives**: Hemiplegic shoulder pain (HSP) is a frequent post-stroke complication affecting 30–65% of survivors, contributing to motor dysfunction and reduced quality of life. Balance impairment is another major concern that increases fall risk. This study aimed to examine the associations between HSP, shoulder range-of-motion (ROM) limitations and balance performance using both clinical and sensor-based evaluations. **Methods:** In this cross-sectional study, 108 stroke survivors (54 with HSP, 54 without) were assessed. Pain intensity was evaluated using the Visual Analog Scale (VAS), balance with the Berg Balance Scale (BBS), and shoulder mobility and postural sway with the validated Euleria Lab IMU-based system integrated with a force platform. Between-group differences were analyzed using the Mann–Whitney U test, and correlations between pain, ROM, balance, and fall-risk indices were determined via Spearman coefficients. **Results**: Participants with HSP had significantly lower BBS scores (20.96 ± 8.71) than those without HSP (34.58 ± 11.71; *p* < 0.001). VAS activity scores were negatively correlated with BBS (r = −0.196, *p* = 0.043) and positively correlated with postural sway and fall-risk parameters, particularly under eyes-closed conditions. Shoulder ROM limitations were linked to poorer balance, and both static and dynamic fall-risk indices were strongly correlated with pain severity (r = 0.676 and r = 0.657; *p* < 0.001). **Conclusions**: HSP was associated with impaired balance and elevated fall risk in stroke survivors. The combination of clinical scales and wearable sensor-based measurements provides a comprehensive understanding of postural control deficits. These findings emphasize the need for rehabilitation strategies targeting pain reduction, shoulder mobility, and balance to support functional recovery.

## 1. Introduction

Stroke remains a leading cause of long-term disability and contributes to substantial functional and socioeconomic burden worldwide [1,2]. Hemiplegia develops in more than 70% of stroke survivors, causing pronounced deficits in mobility and activities of daily living [3,4]. Among post-stroke complications, hemiplegic shoulder pain (HSP) is particularly prevalent, affecting approximately 30–65% of survivors [5,6,7]. The etiology of HSP is multifactorial, including glenohumeral subluxation, rotator cuff pathology, spasticity, and central pain mechanisms [8,9]. Regardless of its origin, persistent HSP is strongly associated with delayed motor recovery, reduced upper-limb use, and diminished quality of life [10]. Balance dysfunction is another major post-stroke issue and a primary determinant of fall risk [11,12]. Impaired postural control often arises from asymmetric weight distribution, altered trunk coordination, and disrupted proprioceptive input. Pain can further contribute to postural asymmetry by limiting shoulder motion, altering scapulothoracic rhythm, and promoting compensatory trunk movements [13,14]. Consequently, HSP may aggravate postural instability and increase the likelihood of falls among stroke survivors. Although both HSP and balance impairment are well-documented consequences of stroke, their interaction remains insufficiently clarified. Previous studies have yielded inconsistent findings—some reporting weak or no associations between shoulder pain and balance [15], while others revealed significantly poorer balance and higher fall risk in patients with HSP [16]. These inconsistencies may be attributed to methodological limitations and the predominant use of subjective or clinician-rated scales. Traditional clinical assessments, such as the Berg Balance Scale (BBS), provide valuable insights but are limited by observer dependency, low sensitivity, and potential ceiling effects. Recent advances in wearable sensor technology have enabled objective, high-resolution quantification of postural sway and movement patterns [17,18,19]. Such systems can capture subtle balance impairments that clinical scales may overlook, offering a more comprehensive understanding of postural control mechanisms in stroke rehabilitation. Therefore, this study aimed to investigate the association between hemiplegic shoulder pain, shoulder range-of-motion limitations, and balance performance using a combined clinical and wearable sensor-based assessment approach. We hypothesized that stroke survivors with HSP would exhibit greater postural sway and reduced balance stability compared with those without HSP.

## 2. Materials and Methods

### 2.1. Study Design and Participants

This cross-sectional observational study was conducted between May and August 2025 in the Department of Physiotherapy and Rehabilitation, Istanbul Aydin University Medicalpark Florya Hospital. Participants were consecutively recruited from the outpatient stroke rehabilitation unit. A priori power analysis (G*Power 3.1) for between-group comparison (Mann–Whitney U test) indicated that 102 participants were required (effect size f = 0.25, α = 0.05, power = 0.80). To account for potential dropouts, 108 participants were included.

A total of 129 stroke patients were screened, and 108 eligible individuals were enrolled based on predefined inclusion and exclusion criteria.

Inclusion criteria: (1) first-ever ischemic stroke confirmed by a neurologist; (2) age 40–80 years; (3) Mini-Mental State Examination (MMSE) score ≥ 23; (4) Brunnstrom stage ≥ 3 for both upper and lower extremities, indicating sufficient voluntary movement for basic ambulation; and (5) willingness to participate.

Exclusion criteria: (1) hemorrhagic or recurrent stroke; (2) severe visual, auditory, or cognitive impairment; (3) orthopedic or neurological comorbidities affecting balance (e.g., peripheral neuropathy, Parkinson’s disease, lower-limb fracture); and (4) significant neglect or communication disorder.

Sample size: An a priori power analysis (G*Power 3.1) for between-group comparison (Mann–Whitney U test) indicated that 102 participants were required to detect a medium effect size (f = 0.25, α = 0.05, power = 0.80). To account for possible data loss, 108 participants were enrolled, which was statistically adequate.

#### Ethical Considerations

Ethical approval for this study was obtained from the Non-Interventional Clinical Research Ethics Committee of Istanbul Aydin University (Approval No: 2025/05, dated 14 May 2025), prior to the commencement of data collection.

### 2.2. Assessment Tools

#### 2.2.1. Pain Assessment

Pain intensity was evaluated using the Visual Analog Scale (VAS). Participants rated their pain on a 10 cm horizontal line anchored with “0 = no pain” and “10 = worst imaginable pain.”

To obtain a comprehensive understanding of pain experience, assessments were conducted under two distinct conditions: at rest and during activity. This approach allowed for the evaluation of both baseline discomfort and movement-induced pain, providing a more nuanced representation of the participants’ pain perception.

The VAS is a valid, reliable, and widely used instrument in stroke rehabilitation research for quantifying subjective pain intensity [11].

#### 2.2.2. Balance Assessment

Balance performance was evaluated with the Berg Balance Scale (BBS), comprising 14 items scored on a 5-point scale, with a maximum of 56 points. Lower scores reflect greater impairment and higher fall risk. The BBS has been extensively validated in stroke rehabilitation [17].

#### 2.2.3. Sensor-Based Assessment

The Euleria Lab system (Euleria Health, Trento, Italy; formerly CoRehab, Riablo™ platform) is a CE Class IIa–certified wearable sensor platform designed for biomechanical and quantitative movement assessment. The system integrates multiple inertial measurement units (IMUs) and a force plate, wirelessly connected to a computer interface for real-time data acquisition.

Each IMU captures three-dimensional acceleration, angular velocity, and orientation data. The manufacturer’s proprietary software processes and filters these signals using a low-pass algorithm to minimize noise and ensure stability. A sensor fusion algorithm combines accelerometer, gyroscope, and magnetometer data to accurately estimate joint angles and postural alignment. Kinematic and kinetic parameters were automatically computed by the Euleria software, including:Center of Pressure (CoP) displacement (mm): total distance traveled by the CoP during the test;Sway velocity (mm/s): average velocity of CoP movement, indicating postural stability;Root Mean Square (RMS) amplitude (mm): variability of sway around the mean CoP position;Symmetry index (%): weight distribution ratio between the affected and unaffected sides;Fall-risk index (score): a composite indicator generated by the system’s algorithm, integrating sway variability and CoP dispersion.

This setup provides objective, high-resolution quantification of postural stability, balance performance, and fall risk in post-stroke individuals.

(a) Shoulder Range of Motion (Pro Module)

In Figure 1, IMU sensors were attached to the dorsal aspect of the wrist to record upper limb kinematics. Before measurement, the system was calibrated in a seated position, with the participant’s thumbs facing forward to ensure neutral alignment of the upper extremity. Both active and passive joint movements—including flexion, extension, abduction, internal rotation, and external rotation—were recorded in real time. Passive measurements were performed by the therapist, ensuring a full but pain-free range of motion. All results were automatically expressed in degrees by the Euleria software, providing a quantitative and reproducible assessment of shoulder mobility.

(b) Balance Assessment (Gait Module)

In Figure 2, Postural stability was assessed using the Euleria Lab system under eyes-open (EO) and eyes-closed (EC) conditions. Participants stood barefoot on the force platform, with IMU sensors placed on the trunk to capture postural sway.

Before testing, each participant was calibrated while standing upright on the board for 20 s, maintaining a neutral posture and facing forward toward a fixed visual target. During data collection, two separate trials were conducted: EO and EC conditions, each lasting 20 s.

Rest intervals were individually adjusted based on patient needs to prevent fatigue. The system automatically calculated root mean square (RMS) sway amplitude, mediolateral (ML) and anteroposterior (AP) deviations, sway velocity, and a symmetry index representing weight distribution. Larger RMS and velocity values indicated reduced balance stability.

The following parameters were automatically generated by the system:Root Mean Square (RMS) sway amplitude—overall magnitude of postural sway;Mediolateral (ML) deviation—average and maximum displacement in the frontal plane;Anteroposterior (AP) deviation—average and maximum displacement in the sagittal plane;Median sway values—central tendency of sway, less affected by outliers.

Larger RMS, ML, or AP deviations indicate impaired balance control. The EO vs. EC comparison provides information about sensory reweighting strategies used to maintain postural stability.

(c) Fall Risk Assessment

In Figure 3, the system’s standardized protocol included sit-to-stand, Timed Up and Go (TUG), half-turn, alternate step, and figure-eight walking tasks. Composite risk scores were generated automatically.

The validity of Euleria Lab outcomes against gold-standard gait analysis has been previously reported [18].

### 2.3. Statistical Analysis

Data analyses were performed using IBM SPSS Statistics version 22.0 (IBM Corp., Armonk, NY, USA). The Shapiro–Wilk test was applied to examine the distribution of variables. Since the data were non-normally distributed, nonparametric tests were used. Mann–Whitney U tests compared continuous variables between groups, and Chi-square tests compared categorical variables. Within-group differences were analyzed using the Wilcoxon signed-rank test.

Correlations between VAS, ROM, BBS, and sensor-based balance parameters were examined using Spearman’s rank correlation coefficients. For all comparisons, effect sizes (r) and 95% confidence intervals (CIs) were calculated to indicate the magnitude and precision of the observed effects.

Normality assumptions were verified before the analyses. Missing data were checked prior to statistical testing, and incomplete cases were excluded listwise. Statistical significance was set at *p* < 0.05.

An a priori power analysis was performed using G*Power 3.1 (medium effect size f = 0.25, significance level α = 0.05, statistical power 1 − β = 0.80). The minimum required sample size was 102 participants. The final sample of 108 participants, approximately 5.9% above this threshold, was therefore statistically adequate.

## 3. Results

A total of 129 post-stroke patients were initially screened for eligibility. Of these, 21 did not meet the inclusion criteria (age > 80 years, severe cognitive impairment, visual or auditory deficits, or comorbid neurological/orthopedic conditions) and were excluded. The final sample comprised 108 hemiplegic patients, who were equally distributed into two groups: those with shoulder pain (Group 1, *n* = 54) and those without shoulder pain (Group 2, *n* = 54). The demographic and descriptive characteristics of both groups are presented in Table 1. No significant between-group differences were found in terms of gender, dominant side, hemiplegic side, smoking status, or lesion site (all *p* > 0.05). Similarly, the distribution of chronic diseases, including hypertension, diabetes mellitus, and heart disease, showed no significant differences between groups (*p* > 0.05). The Chi-square (χ^2^) test was used for categorical variables and the Mann–Whitney U test for continuous variables.

Descriptive statistics for age and disease duration are shown in Table 2. Mean age was 64.25 ± 9.52 years (range: 46–80) in Group 1 and 66.09 ± 9.92 years (range: 45–80) in Group 2. Mean disease duration was 10.04 ± 6.68 months in Group 1 and 10.17 ± 5.99 months in Group 2. Neither age nor disease duration differed significantly between groups (*p* = 0.200 and *p* = 0.650, respectively).

### Clinical Assessments

As shown in Table 3, in Group 1, the Berg Balance Scale (BBS) demonstrated moderate positive correlations with shoulder flexion (r = 0.412, *p* = 0.002) and abduction (r = 0.398, *p* = 0.003). In Group 2, weak positive correlations were observed between BBS and flexion (r = 0.356, *p* = 0.007) and abduction (r = 0.332, *p* = 0.009). No significant correlations were found with other shoulder range of motion parameters (*p* > 0.05).

Scores obtained from clinical evaluation scales are summarized in Table 4. Group 1 patients (with shoulder pain) had mean VAS activity and rest scores of 6.00 ± 1.86 (range: 0–9) and 2.85 ± 1.84 (range: 0–6), respectively. Although Group 1 showed higher VAS scores compared with Group 2, these differences were not statistically significant (*p* > 0.05). Berg Balance Scale (BBS) scores were significantly lower in Group 1 compared with Group 2 (20.96 ± 8.71 vs. 34.58 ± 11.71; *p* < 0.001), indicating poorer balance performance among patients with shoulder pain.

As shown in Table 5, correlation analysis revealed a significant negative correlation between VAS (activity) scores and BBS scores (r = −0.512, *p* < 0.001), as well as between VAS (rest) and BBS (r = −0.476, *p* < 0.001). This indicates that greater shoulder pain intensity was associated with reduced balance ability in post-stroke patients.

In Table 6, for the BBS, moderate negative correlations were found with sway median (r = −0.520, *p* = 0.001) and sagittal maximum sway (r = −0.561, *p* = 0.001), while weak negative correlations were observed with sagittal median (r = −0.278, *p* = 0.004), frontal median (r = −0.331, *p* = 0.001), RMS median (r = −0.352, *p* = 0.001), and frontal maximum sway (r = −0.490, *p* = 0.001).

For VAS activity scores, weak to moderate positive correlations were observed with sagittal maximum sway (r = 0.324, *p* = 0.001), RMS median (r = 0.501, *p* = 0.001), and frontal maximum sway (r = 0.291, *p* = 0.002), whereas very weak correlations were found with sway median (r = 0.018, *p* = 0.854) and sagittal median (r = 0.112, *p* = 0.247).

Under eyes-open (EO) conditions (Table 7), BBS scores demonstrated a weak negative correlation with RMS sway (r = −0.259, *p* = 0.007), while other parameters showed very weak or non-significant associations. Conversely, VAS activity scores exhibited significant positive correlations with multiple EO parameters, including RMS sway (r = 0.547, *p* < 0.001), sagittal median (r = 0.271, *p* = 0.005), sagittal maximum sway (r = 0.293, *p* = 0.002), frontal maximum sway (r = 0.450, *p* < 0.001), and frontal median (r = 0.315, *p* = 0.001).

Correlations between BBS, VAS activity scores, and fall-risk indices are presented in Table 8. For the BBS, weak but significant correlations were observed with static fall risk (r = 0.248, *p* = 0.010) and dynamic fall risk (r = 0.249, *p* = 0.010). Very weak correlations were found with balance (r = 0.301, *p* = 0.135), strength (r = 0.176, *p* = 0.002), and mobility (r = 0.230, *p* = 0.137).

For VAS activity scores, moderate positive correlations were detected with static fall risk (r = 0.676, *p* < 0.001) and dynamic fall risk (r = 0.657, *p* < 0.001). Additional weak correlations were found with balance (r = 0.277, *p* = 0.004) and strength (r = 0.378, *p* = 0.001), while mobility showed a very weak, non-significant correlation (r = 0.133, *p* = 0.169).

## 4. Discussion

This study demonstrated that hemiplegic shoulder pain (HSP) was associated with reduced balance performance and increased fall-risk indices in post-stroke patients. Participants with HSP had significantly lower Berg Balance Scale (BBS) scores and poorer sensor-derived stability parameters than those without shoulder pain. Furthermore, VAS activity scores were significantly correlated with multiple sway metrics and fall-risk indices, indicating that pain severity was associated with impaired postural control and functional safety after stroke [15].

Several biomechanical and neurophysiological mechanisms may explain these associations. Pain in the hemiparetic shoulder often leads to protective postures, muscle guarding, and restricted active range of motion, which disrupt symmetrical weight bearing and trunk alignment [7,9]. In addition, altered proprioceptive feedback from the painful shoulder joint may affect cortical and subcortical sensorimotor integration, reducing postural stability [12]. In our cohort, significant correlations between VAS activity and sway parameters further support this concept, suggesting that higher pain levels are linked to increased postural instability.

The present findings align with previous studies reporting the impact of HSP on postural stability and functional outcomes. Gayretli Atan and Pehlivan (2025) demonstrated that HSP negatively affected balance, upper extremity function, and quality of life in stroke survivors [15]. Similarly, Baierle et al. (2013) found reduced postural stability in individuals with painful shoulder disorders compared with healthy controls [12]. Our sensor-based results are consistent with previous reports indicating increased sway and asymmetry in pain-related postural disturbances [18,19]. Unlike studies relying solely on clinical scales, the integration of Euleria Lab sensor data in this study allowed for more sensitive quantification of postural instability.

Combining clinical and sensor-based assessments provides a comprehensive understanding of balance impairment in post-stroke individuals with HSP. Parameters such as RMS sway, frontal and sagittal deviations, and fall-risk indices can support early detection of instability and objective monitoring of rehabilitation progress [18,20,21]. Furthermore, the integration of wearable sensor systems and biofeedback-based exergaming may enhance motivation, improve adherence, and facilitate individualized rehabilitation planning [20,21].

This study has several important limitations. First, the Riablo system used for sensor-based balance assessment was originally designed for populations over 65 years of age; therefore, the inclusion of participants younger than 65 years may have affected measurement sensitivity. In addition, since the device’s normative values were developed based on the Italian population, obtaining identical results in individuals with different demographic characteristics may represent another limitation. Furthermore, the system’s ability to detect even very small movements may have led to discrepancies with clinical scale outcomes, which primarily reflect macro-level performance.

In addition, the absence of multivariate analyses such as ANCOVA or regression models represents another limitation of this study, as the relatively small and homogeneous sample did not allow for reliable adjustment for potential confounding factors.

Future research should investigate the relationship between HSP and postural control using longitudinal or interventional study designs. Neuroimaging and electrophysiological methods may help to elucidate the neural mechanisms underlying pain-related postural instability. Moreover, the analysis of sensor-based data through machine learning algorithms could enhance fall-risk prediction and contribute to the development of personalized rehabilitation strategies [21,22].

## 5. Conclusions

This study provides preliminary evidence that hemiplegic shoulder pain (HSP) is associated with impaired balance and increased fall risk in stroke survivors. HSP patients showed lower Berg Balance Scale scores and less favorable sensor-derived parameters, with pain severity correlating with measures of instability.

Combining clinical and sensor-based assessments offers initial insight into how shoulder pain and limited mobility may affect overall postural stability. Sensor-based tools showed promise for objectively quantifying HSP-related balance deficits but require further validation in larger cohorts.

Future longitudinal studies are needed to confirm these associations and to determine whether targeted pain management can effectively reduce fall risk.

## Figures and Tables

**Figure 1 sensors-25-07665-f001:**
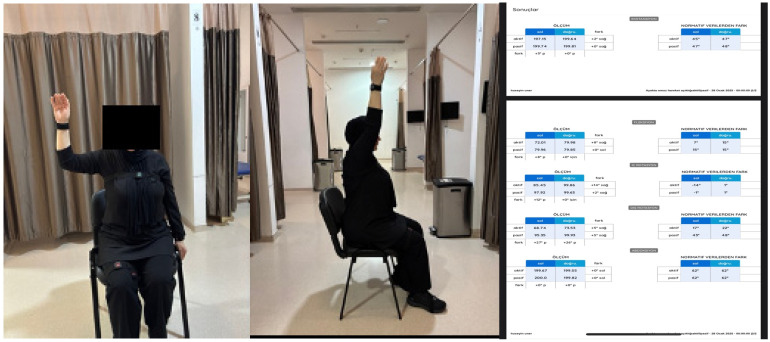
Shoulder range of motion assessment and example results.

**Figure 2 sensors-25-07665-f002:**
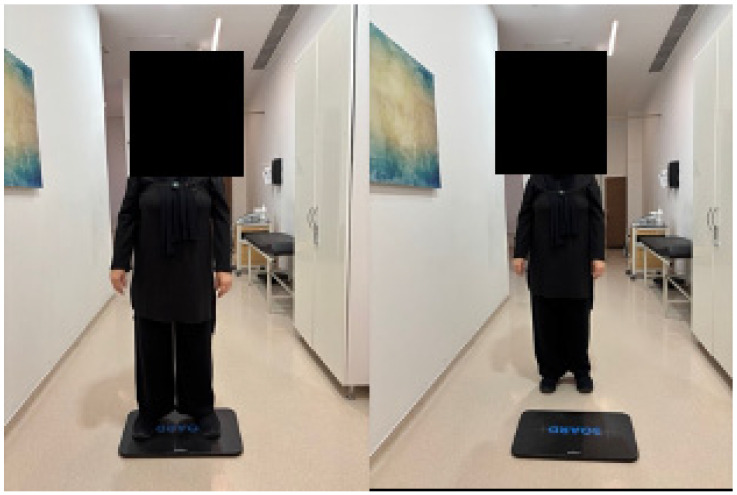
Balance assessment with the Euleria Lab.

**Figure 3 sensors-25-07665-f003:**
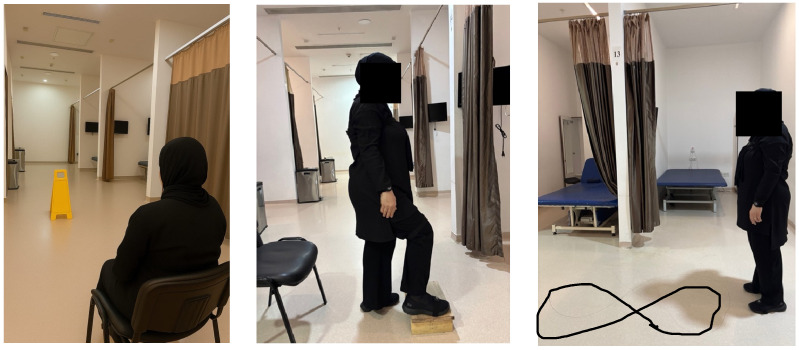
Assessment of balance and fall risk with the Euleria Lab system.

**Table 1 sensors-25-07665-t001:** Demographic and descriptive characteristics of the groups.

	Group 1 N (%)	Group 2 N (%)	*p*
Gender			
Male	32 (59.3)	34 (63)	0.554
Female	22 (40.7)	20 (37)	
Dominant Side			
Right	52 (96.3)	52 (96.3)	0.310
Left	2 (3.7)	2 (3.7)	
Hemiplegic Side			
Right	25 (46.3)	25 (46.3)	0.123
Left	29 (53.7)	29 (53.7)	
Smoking Status			
Yes	14 (25.9)	12 (22.2)	0.654
No	40 (74.1)	42 (77.8)	
Lesion Site			
Cortical	24 (44.4)	17 (31.5)	
Subcortical	19 (35.2)	24 (44.4)	0.378
Cortical + Subcortical	11 (20.4)	13 (24.1)	
Chronic Diseases			
Hypertension	18 (33.3)	16 (29.6)	
Diabetes Mellitus	7 (13)	1 (1.9)	
Hearth Disease	3 (5.6)	0 (0)	
HT + DM	10 (18.5)	14 (25.6)	0.101
HT + Heart Disease	4 (7.4)	3 (5.6)	
DM + Heart Disease	4 (7.4)	4 (7.4)	

**Table 2 sensors-25-07665-t002:** Descriptive statistics of age and disease duration of the groups.

	Group 1 (N = 54) Mean ± SS (Min–Max)	Group 2 (N = 54) Mean ± SS (Min–Max)	*p*
Age (Years)	64.25 ± 9.52 (46–80)	66.09 ± 9.92 (45–80)	0.20
Disease Duration (Months)	10.04 ± 6.68 (3–13)	10.17 ± 5.99 (3–23)	0.650

**Table 3 sensors-25-07665-t003:** Relationship between the Berg Balance Scale and shoulder range of motion parameters between groups.

		Group 1 (N = 54)	Group 2 (N = 54)
Active shoulder flexion	*p*	0.000	0.001
	r	0.467	0.461
Passive shoulder flexion	*p*	0.018	0.110
	r	0.323	0.348
Active shoulder abduction	*p*	0.000	0.003
	r	0.483	0.400
Passive shoulder abduction	*p*	0.004	0.008
	r	0.390	0.361
Active shoulder internal rotation	*p*	0.000	0.012
	r	0.524	0.342
Passive shoulder internal rotation	*p*	0.310	0.245
	r	0.297	0.165
Active shoulder external rotation	*p*	0.001	0.052
	r	0.456	0.268
Passive shoulder external rotation	*p*	0.363	0.004
	r	0.007	0.390

**Table 4 sensors-25-07665-t004:** Scores obtained from the clinical evaluation scales of the groups.

	Group 1 (N = 54) Mean ± SS (Min–Max)	Group 2 (N = 54) Mean ± SS (Min–Max)	*p*
Shoulder VAS Activity Score	6.00 ± 1.86 (0–9)	—	0.00
Shoulder VAS Rest Score	2.85 ± 1.84 (0–6)	—	0.00
Berg Balance Scale Score	20.96 ± 8.71 (6–52)	34.58 ± 11.71 (6–52)	0.928

**Table 5 sensors-25-07665-t005:** Correlation of VAS activity score and Berg Balance Scale.

		VAS Activity Score
Berg Balance Score	*p*	0.043
	r	0.196

**Table 6 sensors-25-07665-t006:** Correlation of balance parameters with the Berg Balance Scale and VAS activity score eyes-closed condition.

		GK Sway Medyan	GK Sagittal Medyan	GK Sagittal Maks	GK Frontal Medyan	GK RMS Medyan	GK Frontal Maks
Berg Balance Scale	r	−0.520	−0.278	−0.561	−0.331	−0.352	−0.490
	*p*	0.001	0.004	0.001	0.001	0.001	0.001
VAS Activity Score	r	0.018	0.112	0.324	−0.550	0.501	0.291
	*p*	0.854	0.247	0.001	0.575	0.001	0.002

**Table 7 sensors-25-07665-t007:** Relationship of the berg balance scale and VAS activity score with balance data under the eyes-open condition.

		GA Sway (RMS)	GA Sagittal Median	GA Sagittal Max	GA Frontal Max	GA Frontal Median
Berg Balance Scale	r	−0.259	−0.112	−0.135	−0.210	−0.137
	*p*	0.007	0.252	0.167	0.300	0.160
VAS Activity Score	r	0.547	0.271	0.293	0.450	0.315
	*p*	0.001	0.005	0.002	0.001	0.001

**Table 8 sensors-25-07665-t008:** Relationship of the berg balance scale and VAS activity score with fall risk and sub-parameters.

		Static Fall Risk	Dynamic Fall Risk	Balance	Strength	Mobility
Berg Balance Scale	r	0.248	0.249	0.301	0.176	0.230
	*p*	0.010	0.010	0.135	0.002	0.137
VAS Activity Score	r	0.676	0.657	0.277	0.378	0.133
	*p*	0.001	0.001	0.004	0.001	0.169

## Data Availability

The data presented in this study are not publicly available due to ethical and privacy restrictions.

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
