# Peer review of "Sensor-Based Assessment of Post-Stroke Shoulder Pain and Balance"

_sensors, 2025, doi:10.3390/s25247665_

Round 1
Reviewer 1 Report
Comments and Suggestions for Authors
- Minor grammatical errors need to be eliminated.
- The author needs to explain the relationship between Hemiplegic Shoulder Pain and Balance Impairment furtherly.
There exists the possibility that other factors may contribute to Balance Impairment.
Author Response
We sincerely thank the reviewers for their valuable and constructive feedback. We have carefully considered all comments and revised the manuscript accordingly. The changes made in response to each point are clearly highlighted in the revised version. We believe that these revisions have substantially improved the quality and clarity of our paper.

Reviewer 2 Report
Comments and Suggestions for Authors
Dear Authors,
Thank you for submitting your manuscript to Sensors. Your study addresses an important and understudied topic: the association between hemiplegic shoulder pain (HSP) and balance deficits after stroke, assessed through a sensor-based system. The integration of objective movement analysis into clinical neurorehabilitation is both novel and relevant. However, the manuscript, as currently presented, requires substantial revision to meet the methodological and reporting standards expected by the journal. Below I provide detailed feedback organized by section.
1. Title
-
The title reflects the study content but could be slightly refined for clarity and conciseness.
Suggested revision: “Sensor-Based Evaluation of Hemiplegic Shoulder Pain and Postural Balance Impairment after Stroke.” -
Ensure inclusion of MeSH-compatible terms such as “Stroke,” “Hemiplegia,” “Postural Balance,” and “Wearable Sensors.”
-
Remove redundant terms (“assessment” and “impairment” overlap conceptually).
2. Abstract and Keywords
-
The abstract correctly follows IMRaD structure but requires more concise reporting.
-
Methods: specify the study design (cross-sectional observational), the number of participants in each group (with and without HSP), and the types of sensors used (e.g., inertial measurement units, force platforms).
-
Results: include quantitative findings (e.g., mean differences, p-values, and correlation coefficients).
-
Conclusions: avoid causal phrasing such as “HSP affects balance”; instead use “HSP was associated with poorer balance performance.”
-
Keywords should be standardized and non-redundant with the title (e.g., “Balance assessment,” “Hemiplegia,” “Postural control,” “Inertial sensors”).
3. Introduction
-
The Introduction effectively outlines the clinical importance of HSP and balance impairment, but it is overly descriptive and lacks a clear statement of novelty.
-
Strengthen the rationale for using sensor-based assessment over traditional clinical scales. Why are objective metrics needed, and what specific gaps in prior research are you addressing?
-
The theoretical framework linking shoulder biomechanics and postural control should be briefly discussed (e.g., altered proprioception, compensatory trunk movements).
-
The aim and hypothesis must be clearly stated in the final paragraph. For example:
“We hypothesized that stroke survivors with hemiplegic shoulder pain would demonstrate greater postural sway and reduced balance stability, as measured by inertial sensors, compared with those without HSP.”
4. Materials and Methods
This section requires greater methodological precision to ensure reproducibility.
-
Design: Explicitly state that this is a cross-sectional observational study.
-
Participants:
-
Clarify recruitment source (rehabilitation center, hospital outpatient service, etc.).
-
Include inclusion/exclusion criteria in full detail (e.g., stroke chronicity, cognitive ability, musculoskeletal comorbidities).
-
Report the sample size calculation or justify the number of participants.
-
-
Instrumentation:
-
Provide a clear description of the sensor system—brand, model, sampling frequency, accuracy, and data processing algorithms.
-
Define all kinematic and kinetic variables (e.g., center of pressure displacement, sway velocity, RMS, symmetry index).
-
Include any filtering or signal processing parameters (e.g., Butterworth filter, cutoff frequency).
-
-
Procedures:
-
Describe the balance test protocol in more detail: stance conditions (eyes open/closed), duration, repetitions, rest periods, and safety precautions.
-
Specify whether participants used footwear or assistive devices.
-
-
Clinical assessments:
-
Include descriptions of scales used (e.g., Modified Ashworth Scale, Fugl-Meyer Assessment, VAS for pain).
-
-
Statistical analysis:
-
The analysis is currently limited to correlations and t-tests. A more robust model (e.g., multivariate regression or ANCOVA) is necessary to adjust for potential confounders (age, sex, chronicity, motor recovery).
-
Report effect sizes and confidence intervals for all comparisons.
-
Verify normality assumptions and describe handling of missing data.
-
5. Results
-
Results are presented clearly but should be reorganized for readability:
-
Participant characteristics (Table 1)
-
Group comparisons (with vs. without HSP)
-
Correlations between pain and balance metrics.
-
-
Include sample size and descriptive statistics for each variable.
-
When reporting correlations, specify the strength and direction using both r and p values.
-
Avoid redundancy between text and tables; summarize key trends in text only.
-
The figures illustrating sway trajectories are informative but need clearer legends (define axes, sampling units, and indicate typical vs. pathological patterns).
6. Discussion
-
The Discussion provides useful context but needs better structure and critical depth.
Suggested structure:-
Principal findings – concise restatement of major results.
-
Interpretation – explanation of biomechanical or neurophysiological mechanisms linking HSP and postural control.
-
Comparison with previous studies – discuss similarities/differences with prior work using both clinical and sensor-based approaches.
-
Clinical implications – highlight how sensor-based tools could improve diagnosis or rehabilitation planning.
-
Limitations – acknowledge the cross-sectional design, small sample, potential selection bias, and unmeasured confounders (e.g., medication, vision).
-
Future directions – suggest longitudinal or intervention studies to test causality.
-
-
Remove speculative or causal language (e.g., “HSP leads to balance deficits”) and use “was associated with.”
-
The discussion should avoid repeating background material already provided in the Introduction.
7. Conclusions
-
The conclusion section should be shorter and more cautious.
-
Avoid overgeneralization. Instead of “sensor-based systems can be used clinically to identify HSP-related balance problems,” use “sensor-based assessment showed promise for objectively quantifying balance deficits associated with HSP, warranting further validation in larger cohorts.”
-
Emphasize the study’s preliminary nature and its contribution to generating hypotheses for future research.
8. Formal and Language Issues
-
The manuscript is generally well written but would benefit from professional English editing to improve fluency and concision.
-
Use consistent terminology (“hemiplegic shoulder pain” instead of alternating with “post-stroke shoulder pain”).
-
Avoid redundancy and long sentences; prefer clear, impersonal phrasing.
-
Check reference formatting for MDPI (Vancouver) style compliance; several DOIs and italics are missing.
-
Figures should include clear units and labels; tables should use consistent decimal precision.
9. Conceptual and Methodological Concerns
-
The study would benefit from a multivariate analytical approach to account for confounders.
-
The potential bidirectional relationship between pain and balance (pain influencing postural control vs. instability aggravating pain) should be explicitly discussed.
-
Consider integrating sensor-derived features (e.g., sample entropy, frequency-domain analysis) that may capture subtle balance impairments.
-
The clinical translation of the sensor system should be more thoroughly discussed—feasibility, cost, and potential for clinical integration.
Overall Assessment
This is a promising and relevant study that integrates sensor technology into neurorehabilitation assessment. However, the current version falls short of publication standards due to insufficient analytical rigor, limited discussion of limitations, and overly assertive conclusions. Strengthening the statistical analysis, refining the structure, and improving the clarity of writing would significantly enhance the manuscript’s quality.
I encourage the authors to undertake a major revision and resubmit after addressing all methodological and interpretive issues.
Sincerely :)
Author Response
We sincerely thank you for carefully evaluating our manuscript and providing detailed, constructive feedback section by section. Your valuable suggestions and insights have greatly contributed to improving the quality of our work. All comments have been carefully reviewed, and the necessary revisions have been made accordingly.

Reviewer 3 Report
Comments and Suggestions for Authors
The manuscript addresses a current and clinically relevant issue concerning the impact of shoulder pain in hemiplegic patients on balance control and fall risk. The authors aimed to analyze this relationship through the application of standardized clinical scales combined with modern a sensor-based technology, which stands out as one of the main strengths of the paper. Starting from the notion that hemiplegic shoulder pain should not be viewed as an isolated musculoskeletal problem, but rather as a potential systemic factor that can affect postural control and overall functional capacity after stroke; the authors appropriately position their study within the existing body of literature.
The methodological approach includes the use of clinical scales such as VAS and BBS, in combination with objective measurements obtained through the Euleria Lab sensor system. This integration of subjective and objective indicators represents good practice in modern neurorehabilitation and enables a more detailed and precise analysis.
The results show that patients with shoulder pain achieve significantly lower scores on the balance assessment scale and display higher values of sway parameters in the sensor-based measurements. It is particularly noteworthy that the differences between groups are more pronounced under eyes-closed conditions, highlighting the important role of impaired proprioception in postural stability regulation. The correlations between pain and instability parameters are statistically significant and confirm the clinical relevance of the observed association.
Although the topic is well chosen and the analysis thoughtfully carried out, there are several aspects of the manuscript that could be further improved:
- There is a lack of transparency in the description of how the sensor data were processed. Although the measured parameters are listed, there is no detailed explanation of how the raw signals were treated. For example, whether specific filters were applied, which measurement units were used, how long the analyses lasted, or how the composite fall risk index was calculated. These details are essential for understanding the depth of the analysis and ensuring the reproducibility of the findings.
- Although Figures 1 and 2 are mentioned in the text, they do not contain clear illustrations that would help the reader understand the measurement setup, sensor placement, or a typical output from the device. Given that the study heavily relies on a technological platform, the inclusion of visual elements would significantly contribute to better understanding of the procedure and interpretation of the data.
- The discussion lacks deeper analysis and critical reflection. A comparison with similar studies (especially those that used different sensor systems or focused on the same target population) would be expected. Furthermore, the clinical relevance of the observed differences and correlations is not sufficiently discussed, nor are the limitations related to the validity and sensitivity of the applied measures addressed.
- The paper does not include data on potential confounding factors, such as medication use, the presence of psychological conditions, or the level of spasticity, all of which may influence both pain perception and balance control. Although the groups are demographically balanced, these factors should at least be mentioned in the text and considered as potential limitations.
Author Response
Thank you so much for your valuable feedback. Your comments mean a lot to me, and I will carefully consider them while revising and improving the manuscript

Round 2
Reviewer 2 Report
Comments and Suggestions for Authors
Dear Authors,
The study is clinically relevant and employs innovative sensor technology to assess a common post-stroke complication. However, major revisions are required to ensure methodological rigor, reporting clarity, and linguistic accuracy. Priority improvements include:
-
Verifying and correcting statistical inconsistencies.
-
Condensing technical descriptions while enhancing methodological transparency.
-
Reducing redundancy across text, tables, and discussion.
-
Refining the English language and adherence to Sensors editorial standards.
Title:
-
The title is informative and relevant but exceeds optimal length; consider shortening to fewer than 12 words.
-
Avoid hyphenation breaks (“Pos- tural”).
-
Include MeSH terms such as Stroke, Hemiplegic Shoulder Pain, Balance, and Wearable Sensors.
Abstract and Keywords:
-
The abstract presents a clear overview but lacks a structured IMRyC format.
-
Key methodological elements (design, tools, variables, and main statistical outcomes) should be explicitly summarized.
-
Results must include concrete numerical outcomes (p-values, r coefficients, effect sizes).
-
Conclusions contain minor interpretative overreach; restrict statements to data-supported findings.
-
Keywords should match MeSH terminology and avoid redundancy with title terms.
Introduction:
-
Provides comprehensive background but is overly long and occasionally repetitive. Condense to approximately 450–500 words.
-
The research gap is identified but buried under general statements; highlight more explicitly why sensor-based assessment provides added value beyond clinical scales.
-
Ensure clear logical flow: stroke → hemiplegia → HSP → balance dysfunction → rationale for sensor-based assessment.
-
Hypotheses are appropriate but should be distinctly stated at the end of the section.
-
Citations are mostly relevant but some are outdated or non-English; prioritize high-quality recent studies (last 5 years).
Materials and Methods:
-
Design: Clearly identify the study as “cross-sectional observational.” Specify whether participants were consecutively recruited.
-
Participants: Inclusion/exclusion criteria are detailed, but redundancy should be reduced. The sample size justification using GPower is appropriate but should specify the targeted statistical test (e.g., between-group comparison).
-
Instruments: The technical description of the Euleria Lab system is excessive for a clinical paper and could be summarized. More critical is to describe how the devices were applied and calibrated for shoulder ROM and postural balance tasks.
-
Data acquisition: Indicate duration, rest intervals, and whether trials were averaged.
-
Statistical analysis: The methods section repeats the power analysis twice. Define dependent and independent variables explicitly.
-
Clarify handling of outliers and missing data.
-
Indicate whether any correction for multiple testing (e.g., Bonferroni) was applied.
-
Justify why correlations and nonparametric tests were preferred and ensure reporting of effect sizes and confidence intervals.
-
Overall, this section demonstrates effort and structure but needs precision, reduction of technical verbosity, and greater focus on reproducibility.
Kind regards
